# Tolerability of 2 and 4 mg/kg Dosing Every 12 Hour of a Cannabidiol- and Cannabidiolic Acid-Rich Hemp Extract on Mixed-Breed Dogs Utilized for Teaching in a Closed Colony

**DOI:** 10.3390/ani14131863

**Published:** 2024-06-24

**Authors:** Trista Mills, Stephanie Myers, Daniel Hughes, Joseph Wakshlag

**Affiliations:** 1School of Veterinary Medicine, Texas Tech University, Amarillo, TX 79106, USA; trista.mills@ttu.edu (T.M.); stephanie.myers@ttu.edu (S.M.); 2Ellevet Sciences, Portland, ME 04106, USA; daniel.hughes@ellevetsciences.com; 3Department of Clinical Sciences, College of Veterinary Medicine, Cornell University, Ithaca, NY 14850, USA

**Keywords:** dog, cannabidiol, cannabidiolic acid, adverse events, tolerability

## Abstract

**Simple Summary:**

Considering the increasing popularity of cannabinoid-rich hemp-based full-spectrum nutraceuticals, there is a need to better understand the tolerability and adverse events associated with their use. This study aimed to examine the effects on dogs when using a 2 mg/kg and 4 mg/kg dose of CBD/CBDA-rich hemp product every 12 h in an academic teaching colony of dogs when compared to the consumption of a placebo soft-gel capsule every 12 h for two weeks. The results indicated that although there were serum concentrations of CBD in the 30–100 ng/mL concentration and 100–200 ng/mL concentration of CBDA, there were no adverse effects or behavioral changes observed regardless of the treatment used, thus showing that in normal healthy dogs, 2–4 mg/kg every 12 h appears to a safe dosing regimen.

**Abstract:**

With the increase in popularity of utilizing cannabidiol (CBD) for human ailments, owners are actively interested in the possible utilization of cannabinoid products for their pets. The evaluation of CBD-rich hemp as an anti-anxiolytic, anti-inflammatory, immunomodulator, and anti-epileptic supplement has been assessed in previous studies in dogs, with adverse events such as ataxia or lethargy noted. In this study, the utilization of CBD-rich hemp was assessed at two concentrations to ascertain the impact on behavior as well as the tolerability of the medication given in a typically recommended dose and then twice that dose. Eighteen dogs were utilized in a randomized, double-blinded, placebo-controlled, 3 × 3 designed study. Each group of six dogs was provided placebo, 2 mg/kg, and 4 mg/kg of a cannabidiol/cannabidiolic acid (CBD/CBDA)-rich hemp in two-week intervals with one-week washout periods between each treatment period. Throughout the 10-week treatment period, student evaluations were performed, simulating clients’ subjective assessments. Improvements in anxiety-related behavior and adverse events related to lethargy and ataxia were not observed and may indicate that the utilization of CBD-rich hemp products for behavioral changes may require higher dosing to mitigate unwanted behaviors in normal, healthy dogs. Furthermore, serum chemistry and serum cortisol were evaluated after each treatment period showing only a mildly significant increase in serum alkaline phosphatase when dosing at 4 mg/kg every 12 h, which is consistent with previously reported CBD dosing at these higher concentrations. Adverse events associated with CBD/CBDA-rich hemp extract given at 2 and 4 mg/kg every 12 h for two weeks were not reported, suggesting that using CBD-rich hemp in young, healthy dogs was safe during two weeks of treatment.

## 1. Introduction

The use of *Cannabis sativa* hemp strains for cannabinoid treatment has gained popularity among pet owners and veterinarians alike over the past 10 years [1,2]. Cannabis Sativa in the United States that has less than 0.3% delta 9-tetrahydrocannabinol (THC) is considered hemp, and is federally legal, creating a burgeoning market of pet supplements made from these low THC cultivars [3]. Current research supports the potential use of cannabidiol (CBD) isolates and CBD/cannabidiolic acid-rich (CBDA) hemp extracts clinically in canine veterinary medicine for ailments including osteoarthritis, refractory idiopathic epilepsy and atopic dermatitis [4,5,6,7,8,9,10]. Currently, clinically relevant dosing has been established at approximately 2–2.5 mg/kg every 12 h generally for osteoarthritis and 2–4.5 mg/kg every 12 h for refractory seizure activity [4,5,6,7,8,9,10]. In general, these are considered safe doses based on 12–36 weeks safety studies [11,12,13,14], and older literature suggesting that even higher doses are potentially safe in dogs [15]. Though safe, there are still questions about adverse events with the continued use of CBD isolates or CBD-rich hemp extracts due to slightly higher doses causing potential lethargy, ataxia, somnolence or appetite changes at doses from 2 to 5 mg/kg either SID or BID [4,5,6,7,8,9,10]. Clinical veterinary assessments of CBD-rich hemp extract have reported physiological issues observed by veterinarians including mydriasis and mildly delayed hopping responses at doses 5–10 mg/kg as a single daily dose within 2 h of ingestion, yet there appears to be tolerance with chronic dosing of 5 mg/kg daily [16].

Behavioral issues or owner-perceived side effects have been observed in the 2–4.5 mg/kg every 12 h range, with some evidence that CBD-rich hemp can be anxietolytic or can possible hinder aggression [17,18,19]. A study assessing aggressive tendencies, using an approximate dose of 2–4 mg/kg daily, showed that over time there was a decrease in aggressive tendencies during 4 weeks of kenneling; however, when compared to placebo there were no significant differences [17]. Another study suggested that 1.4 mg/kg of CBD 4–6 h prior to a recorded thunderstorm event in shelter/teaching dogs at a University showed no behavioral changes related to this dose of CBD isolate after dosing [18]. In kenneled dog studies on separation and car ride anxiety models, utilizing 4 mg/kg of CBD isolate as a one-time dose, there was found to be minimal physiological alteration in parameters associated with anxiety within 2 h of dosing (i.e., cortisol, lip-licking or heart-rate variability), yet behaviorist video evaluations suggest a more relaxed demeanor [19]. To date, no clinical study in client-owned populations have been performed related to behavioral pathologies.

We hypothesized that increasing the dosage from 2 mg/kg every 12 h to 4 mg/kg every 12 h would elicit adverse effects, behavioral alterations resulting in tolerability issues. The aim of our study was to examine adverse events at two different doses of CBD/CBDA-rich hemp within the dosing range utilized in multiple studies using medical records and weekly veterinary examinations for adverse events; as well as student evaluations of the dogs in an “owner-like” unvalidated observation related to their kenneled lifestyle. A second objective was to determine the serum chemistry alteration that may be attributed to dosing of 2 mg/kg and 4 mg/kg every 12 h for two weeks in a randomized placebo blinded cross-over design to better understand tolerability.

## 2. Materials and Methods

### 2.1. Dogs

The study population included 20 dogs acquired from the local shelter chosen over a period of two days. Dogs that had “good temperament” and were considered “people friendly” were chosen. A 3-week acclimation period to the long-term teaching colony was observed for all dogs. During this time, unsterilized animals were spayed or neutered, screened for parasites and rickettsial diseases, and ancillary procedures necessary for optimal quality of life were performed. Dogs were allowed to undergo these procedures and were included in the study after these medical screens were deemed negative and/or treated appropriately. All dog’s received a complete blood count and a serum biochemistry prior to enrollment and were deemed free of any clinically significant organ-related dysfunction with a physical examination showing all physiological parameters within normal limits. The study was performed with approval from the Texas Tech University School of Veterinary Medicine Institutional Care and Use Committee approval (protocol 2022-1272).

### 2.2. Randomization, Treatment and Physical Assessment

Twenty dogs were randomized, in a double-blinded, placebo-controlled, study design with one week washout periods between treatment periods (two week duration) and each dog was assigned to a sequence of treatment (i.e., ACB, BAC, or CBA). The cohort was randomized into two groups of 7 and one group of 6 dogs, each ranging in size from 10 to 40 kg, using a random number generator (randomizer app version 8.0). All resources for production including the hemp extract, sesame oil, and gelatin were food grade products made under certified and audited good manufacturing processes. Dogs were provided oral soft gel capsules (gelatin based) in groups A (2 mg/kg CBD/CBDA), B (placebo-sunflower oil), and C (4 mg/kg CBD/CBDA) using incremental dosing based on weight of the dog at the beginning of the study phase using capsules designed at 5 mg, 10 mg or 20 mg of total cannabinoid. Each ml of the oil utilized to fill the gel capsules was 29 mg/mL CBD, 30 mL CBDA, 1.2 mg of THC, 1.4 mg/mL tetrahydrocannabinolic acid (THCA), 0.9 mg/mL of cannabichromene (CBC), 1.0 mg/mL cannabichromenic acid (CBCA), 1.2 mg/mL cannabigerol (CBG), 1.2 mg/mL cannabigerolic acid (CBGA) and no other detectable cannabinoids based on third party analysis in a 17025 certified facility (Proverde Analytics, Milford, MA, USA). The hemp oil was also certified free of detectable nonpathogenic or pathogenic microbes, heavy metals, pesticides, extraction solvents and mycotoxins at the same certified facility. Treatment was provided every morning at approximately 7 am and in the evening at approximately 6 pm in a “pill pocket” with the dog’s meals; dogs were weighed every 3 weeks and dose was adjusted appropriately. There was a 1-week washout period and then each group transitioned to the next treatment formulation based on the prior literature that behavioral effects are transient and that any residual cannabinoid in the serum would be negligible to final cannabinoid assessment after the 1-week washout and then two weeks of treatment [20]. To assess physiologic response to the treatments, blood was taken before and after each treatment period (6 samples total) on all dogs in the study to assess cannabinoid concentrations, cortisol, and serum biochemistry, with complete blood counts performed at the initiation and end of the 9-week study period. These samples were taken between 3 and 4 h after morning treatments to provide consistency for serum cannabinoid testing. Physical examinations were performed by the veterinary principle investigator (TM) every week where the dogs were examined and received neurological (i.e., cranial nerve, reflexes, ataxia assessment and proprioceptive testing) and orthopedic examinations (i.e., gait, lameness and proprioception).

### 2.3. Student Interaction Evaluations and Video Assessment

Two weeks before starting the study, students in their first year of the veterinary curriculum worked with their specifically assigned dog for teaching clinical skills examinations. Dogs were then started on the protocol as described and visualized in Figure 1, with both students and veterinarians blinded to treatments. Basic behavior evaluations assessing subjective anxiety, lethargy, ease of working with the dog, aggressive behaviors, jumping behavior, and vocalization/barking were tracked utilizing a Likert scale administered via online survey for the students. Students were asked to fill out evaluations at each interaction with the dog over the 9-week period (Appendix A). Scaling was based on 0 suggesting no changes, scores of −1 being a little worse, −2 being a lot worse, 1 being a little better and 2 being a lot better.

Evaluations were assessed as a change from baseline after interacting with the dogs for two weeks (week 0) for each student interacting with each dog only if they had an evaluation between days 4 and 14 of treatment in all three treatment periods. Typically, for each dog there were 4–6 complete student evaluations for each treatment period which were averaged across the students for each category of behavior/adverse event assessment and presented as a mean and standard deviation score for that specific dog during each treatment period.

A secondary behavior assessment was performed looking at panting, circling, self-mutilation/licking, posture in kennel, sleep habits, interest in enrichment, utilizing objective video recording performed once weekly between 6 and 7 pm for a 10 min duration approximately 1 h after evening dosing, which was recorded and assessed via remote observation. A running medical evaluation kept track of any illness experienced by the dogs through laboratory animal care attendants and veterinarians’ observations for any illness including attitude changes, decreased appetite, vomiting, diarrhea, coughing, or other clinical signs related to illness or injury.

#### 2.3.1. CBC, Serum Chemistry and Cannabinoid Analysis

Blood was drawn from each dog prior to and after the completion of each phase of treatment, the day before the treatment started and then again on day 14 of treatment. Two cc of blood was collected in a 4 mL EDTA tube (Beckton Dickenson, Franklin Lakes, NY, USA) while 4 cc was collected in a red top tube that was allowed to clot for 20–40 min before being spun at 3600 G for 10 min to collect serum with 1 cc allocated to clinical chemistry for serum biochemical and cortisol assessment. A 1 mL aliquot of serum was also frozen at −80 °C for eventual cannabinoid analysis. CBCs were performed at week 0 and week 9 of the study to assess any hemogram changes from initiation to study end (IDEXX Laboratories, Memphis, TN, USA). CBC parameters assessed were red blood cell count, hemoglobin, hematocrit, white blood cell count, neutrophils, lymphocytes, eosinophils, basophils and platelets. Serum biochemistry and cortisol assessment was performed prior to and at the completion of each phase of the experiment (IDEXX Laboratories, Memphis, TN, USA). Parameters assessed through serum chemistry included glucose, symmetric dimethyl arginine, serum urea nitrogen (SUN), creatinine, sodium, potassium, chloride, magnesium, calcium, phosphorus, total protein, globulin, albumin, alkaline phosphatase (ALP), alanine aminotransferase (ALT), aspartate aminotransferase (AST), bilirubin, cholesterol and cortisol concentrations.

At study end, all frozen samples were shipped on dry ice overnight to Cultivate Analytics (Portland, ME, USA) and were assessed utilizing liquid chromatography and mass spectroscopy (LC/MS/MS) for serum cannabinoids. Serum assessment included CBD, CBDA, THC, THCA, CBC, CBCA, CBGA, 6-hydroxy cannabidiol (6-OH-CBD), 7 hydroxy-cannabidiol (7-OH-CBD), 7 carboxy-cannabidiol (7-COOH-CBD), cannabinol (CBN) and glucoronidated-THC.

Analysis was performed with a fit-for-purpose LC/MS/MS method for measurement of seven cannabinoids and five of their metabolites at Cultivate (Portland, ME, USA). Matrix-matched calibration curves were generated for each compound using reference standards obtained from Cerilliant Corporation (Round Rock, TX, USA). An internal standard solution was generated in methanol at 1 µg/mL containing 7-OH-CBD-d3, 11-OH-delta-9-THC-d3, CBDA-d3, CBD-d3, delta 9-THC-d3, and THCA-A-d3. Dog serum samples were prepared for analysis by adding 100 µL of serum to 250 µL of methanol and 50 µL of internal standard solution. Samples were vortexed for at least 1 min and centrifuged at 10,000 rpm (9300× *g*) at 4 °C. The supernatant was transferred to autosampler vials for analysis via LC-MS/MS (Agilent 1260 Infinity II HPLC coupled to an Agilent 6490 Triple Quadrupole; Santa Clara, CA, USA). A volume of 10 µL of each sample was injected onto an Agilent column (Poroshell 120 EC-C18 2.7 µm 3.0 × 100 mm). The columns were equilibrated with 50% mobile phase A (0.1% formic acid in water) and 50% mobile phase B (0.1% formic acid in acetonitrile). The column compartment was held at 50 °C. The compounds were eluted by a starting gradient of 50% B held for 0.5 min, ramped to 100% B over 7 min, and held at 100% B for 1 min. The column was re-equilibrated to the initial mobile phase composition for 1.5 min. The flow rate was 0.8 mL/min for the entire analysis. The compounds were detected in electrospray ionization positive and/or negative mode as described in Appendix A. The gas temperature and sheath gas temperature were set to 120 °C and 400 °C, respectively. The gas flow and sheath gas flow were set to 11 L/min and 12 L/min, respectively. The nebulizer was set to 30 psi. The capillary and nozzle voltage were set to 3000 V and 2000 V, respectively. Concentrations were calculated using Agilent Mass Hunter Quantitative Analysis version 10.0 using a linear regression with 1/c weighing based on relative response for each compound. The lower limit of detection and quantitation in dog serum is provided in Appendix A.

#### 2.3.2. Statistical Analysis

##### Student Adverse Event/Behavioral Evaluations

The means from student evaluations from each treatment phase as changes from baseline were assessed for normality utilizing Shapiro–Wilk test, and residual plots were examined. If normality was rejected, the data were log transformed and visually inspected for normal distribution before analysis, utilizing repeated measures mixed-model analysis on JMP 16 (Wittington House, UK). Variables included in the model as fixed effects were gender, treatment, time, sequence, and trazodone use. Random effects were period and dog nested within period. Tukey’s post hoc tests were performed for multiple comparison correction of *p*-values for all pairwise least square means (LSM) comparisons found to be significant, with a value of 0.05 or less being deemed a significant finding.

#### 2.3.3. CBC and Serum Chemistry

All CBC and serum chemistry data were assessed utilizing Shapiro–Wilk test for normality, and residual plots were examined. When normality was rejected, the data was log transformed and visually inspected for normal distribution before analysis utilizing repeated measures mixed-model analysis on JMP 16 (Wittington House, UK). Different covariance structures were tested for each model and the data was then back transformed for reporting if logarithmic analysis was utilized. For the serum chemistry, the variables included in the model as fixed effects were gender, sequence of treatment, time, treatment, and the interaction term treatment × time. Random effects were period and dog nested within period. For CBC pre- and post-trial results, the variables included in the model were gender, sequence of treatment, time with the random effects of period and dog nested within period. Tukey’s post hoc tests were performed for multiple comparison correction of *p*-values for all pairwise LSM comparisons found to be significant, with a value of 0.05 or less being deemed a statistically significant finding.

#### 2.3.4. Serum Cannabinoids

Serum cannabinoids that were detectable within the LLOQ of detection were assessed using descriptive statistics to provide serum concentrations found during each treatment phase. Serum was collected prior to the initiation of each phase and then again between 3 and 4 h after treatment on day 14 of each group. Descriptive statistics including means (medians and ranges) for the two treatments are reported to examine relative cannabinoids and cannabinoid metabolites observed in this population of dogs.

## 3. Results

### 3.1. Physical Examinations

During the 9-week protocol, there were two dogs that were returned to the shelter due to inability to control inter-dog aggression at week 4 of the protocol, leaving 16 dogs for complete data assessment. Demographics including presumed predominant breed, age, gender and weights over the trial are reported in Table 1. Each weekly physical examination of each dog revealed no abnormalities including physical examination, neurologic (deficits in cranial nerves, reflexes, ataxia, or proprioception/hopping responses) or orthopedic examination (gait or lameness evaluations) at any time point. All dogs showed normal appetite throughout the protocol with the average weight of the dogs being 23.7 ± 6.7 kg prior to the trial and 24.1 ± 6.6 kg at the end of the trial. During the protocol, there were two incidences of diarrhea which were treated with metronidazole and probiotic for a 5-day duration; one being during the treatment with 2 mg/kg twice daily and another dog during the treatment with placebo. No other adverse events were noted during the treatment or washout periods that could be attributed to the treatments provided.

Due to kennel management issues, some dogs were placed on trazadone treatment at 5 mg/kg body weight due to clinical assessments by handlers in managing dogs through the recommendation of the clinical vets from the laboratory animal veterinary care services. There were 6 dogs treated with approximately 5 mg/kg every q 12 h trazodone during the placebo treatment, 4 dogs during the 2 mg/kg CBD-rich hemp treatment, and 6 dogs during the 4 mg/kg CBD-rich hemp treatment. Treatment was deemed necessary for exuberant behavior and easier handling of the dogs by students who were performing required enrichment and for teaching purposes related to physical examination as part of their training.

### 3.2. Student Adverse Event/Behavioral Evaluations

At the beginning of the trial, there were 7–8 students assigned to each dog to partake in evaluations during each phase of the trial. After examining all students that responded during each phase of the trial periods, there were between 4 and 6 evaluations from students that had an evaluation for each phase of the study for their respective dog. The average response score from each student over the three phases was averaged for each treatment period and reported as the mean and standard deviation (Table 2). All data collected were normally distributed, so no transformations of the data were required before statistical analysis. Across all the evaluations, there were no differences found in treatment, gender, sequence or trazodone treatment for lethargy, anxiety, jumping, mouthing/biting, or vocalization. A significance was found for handling and treatment effects (*p* = 0.008), whereby post hoc assessment showed improved handling between the 2 mg/kg and 4 mg/kg treatment groups (*p* = 0.004), but not with the placebo. Additionally, the sequence was significantly different for handling (*p* = 0.04) with this difference being between the sequence of 2 mg/kg–4 mg/kg–placebo vs. 4 mg/kg–placebo–2 mg/kg for biting/mouthing behaviors. It was found that there was a significant increase in vocalization during period 2 when compared to period 3 (*p* = 0.02).

### 3.3. Video Evaluation

Ten-minute evaluations of each dog were recorded to assess kennel behaviors including pacing, cage biting, kennel positioning, and vocalization as markers of stress and anxiety in the kenneled environment between 7 p.m. and 7:30 p.m. on day 7 of treatment and at the end of each phase of testing on day 14. Video recordings of all 16 dogs that remained in the study for the duration showed that 15 of the 16 dogs were all recumbent on their sides or in a curled posture primarily sleeping for the duration of each video assessment. There was one dog that was observed to be vocalizing during each evaluation recording, regardless of the treatment period and was persistently vocalizing at the front of the cage for the entire 10 min time period.

### 3.4. Complete Blood Count

Of the CBC results analyzed, all the data was found to be normally distributed. Blood counts assessed at week 0 and week 9 of the study revealed time-related differences in a few parameters (Table 3). There was a significant decrease in reticulocyte across the population of dogs, yet all remained within the reference range (*p* < 0.001). Hematocrit and hemoglobin concentrations were also significantly higher at the end of the study (*p* = 0.004 and *p* = 0.003, respectively). Overall, white blood cell and neutrophil counts were significantly lower at the end of the study as well, (*p* = 0.02 and *p* = 0.015 respectively) with only one dog being above the reference range for white blood cells and neutrophils at the beginning of the study period which normalized at the end of the study. The basophil counts were also significantly decreased from baseline by the end of the study, with all dogs being within reference range at each time point (*p* = 0.015).

### 3.5. Serum Biochemistry and Cortisol

Of the serum biochemistry evaluated, only ALP and cortisol were found to be non-normally distributed and log transformation was performed before evaluation and back transformed for reporting purposes (Table 4). Serum mineral and electrolyte changes were not observed as treatment effects; however, there was a time effect on both serum potassium and serum calcium concentrations. Calcium was typically higher pretreatment for each group than post-treatment (*p* < 0.001), while serum potassium was slightly higher in the post-treatment intervals than prior to treatments (*p* < 0.001). Additionally, serum glucose showed mild significant differences over time (*p* < 0.001), with pretreatment samples being slightly higher than post-treatment samples. Serum proteins including total protein and albumin were significantly higher in males than in females, yet all parameters fell within normal reference ranges (*p* = 0.02 and 0.01, respectively). All serum hepatic enzymes and renal parameters showed no differences regardless of time or treatment except ALP values which showed a treatment*time effect (*p* = 0.001), with post hoc analysis showing the 4 mg/kg every 12 h ALP was higher with 3 of 16 values above the reference range at the end of the treatment period (Figure 2). Serum cortisol showed no significant differences over the treatment period or between treatment groups. Both bilirubin and ALT were elevated in the group designated to the CBA series of treatment compared to the other two randomization sequences (*p* = 0.002 and *p* = 0.01, respectively).

### 3.6. Cannabinoid and CBD Metabolite Concentrations

Detectable cannabinoids and metabolites in the serum were CBD, CBDA, 9-THC, THCA, CBCA, 6-hydroxy cannabidiol (6 OH-CBD), and 7 carboxy cannabidiol (7 COOH-CBD) above their lower limits of quantitation in the 2 mg/kg and 4 mg/kg treatment groups (Table 5). All other cannabinoids were non-detectable in the serum of the dogs. Serum samples collected after each one-week wash out showed no residual cannabinoids above the LLOQ for any of the cannabinoids or metabolites. The analysis of the CBD concentrations on the last day of each treatment period, performed 3–4 h after the morning dose, had the expected results for Group A (placebo) with no detectable cannabinoids for all animals involved. Group B’s (2 mg/kg) post-treatment analysis can be found in Table 5. For metabolites 6-OH-CBD and 7-COOH-CBD at 2 mg/kg, nearly all dogs (>12) had values below the LLOQ and are thus not reported. At 4 mg/kg dosing, four dogs were in between the LLOQ and LLOD; therefore, the values utilized for calculation were half of the LLOQ at 2.5 ng/mL for 6OH-CBD. The 7-COOH-CBD concentrations at 4 mg/kg revealed eight dogs with values between the LLOD and LLOQ; therefore, half of the LLOQ being 0.5 mg/mL was entered as data points for those eight dogs to provide the mean, median and range for this metabolite.

## 4. Discussion

The use of CBD-rich hemp has been observed to induce transient clinical signs of a delayed hopping response and mydriasis in one clinical study at 5 mg/kg, which, when dosed over 5 days diminished with repeated 5 mg/kg daily dosing. Based on veterinary examinations weekly, this was not observed at the end of week one or week two of our study period, suggesting that acclimation to the dose may be occurring or that this unique hemp blend did not induce these physical examination changes. No other clinical signs or observations by veterinarians or students suggested any adverse events related to every 12 h treatment with 2 or 4 mg/kg. In clinical studies utilizing CBD/CBDA-rich hemp extract to determine effect on pruritis, Loewinger reported a portion of the study participants (5/17 dogs) experienced lethargy or calmness attributed to the treatment with no other clinically notable adverse effects when utilizing 2 mg/kg q 12 h [7]. While studies utilizing CBD concurrently with traditional seizure medication (singular or combination use of phenobarbital, zonisamide, levetiracetam and potassium bromide), McGrath and colleagues noted adverse effects including somnolence and ataxia with the addition of 2.5 mg/kg to every 12 h dosing of a CBD product to established treatment protocols for epilepsy [9]. This agrees with the findings reported by Garcia et al. as 3/14 dogs utilized in their study developed transient ataxia during the course of treatment with 2 mg/kg twice daily dosing of CBD/CBDA-rich hemp added to an established treatment protocol of 2 or more anti-seizure medications [8]. Both of these studies note that anti-seizure medications and CBD undergo metabolism via hepatic and gastrointestinal cytochrome p450 and hepatic conjugation activities which could alter the serum concentrations of typical anti-epileptic drugs; however, in the Garcia study, no serum alterations of phenobarbital or zonisamide were observed [8,9,21]. In a later study by Donovan et al., the interactions of CBD and phenobarbital were elegantly examined showing no influence of CBD on phenobarbital serum concentrations in normal healthy beagles at similar or higher dosing [22].

In our study, we did not see any notable alterations in CBC parameters that were related to a treatment effect; however, CBCs were not performed at each stage of the study. Repeated CBCs at each phase were not performed due to prior studies showing no alterations in any CBC parameters using doses between 5 and 10 mg/kg per day in studies examining mixed cannabinoids or CBD-rich hemp products, to date [11,21]. All of the changes over time surrounded primarily red blood cell alterations and reticulocytosis which could be related to environment change as feral dogs and treatment of GI parasitism. More importantly, these changes were not clinically significant as all dogs fell within the normal ranges for RBC and HCT.

Serum chemistry was followed for each phase of treatment as multiple studies utilizing between 2 and 10 mg/kg total daily dose of various CBD products have shown alterations in ALP, with hints towards ALT rises when providing 10 mg/kg as a single daily dose in dogs [4,9,10,12,13,14]. An ALP increase is attributed to the upregulation of cytochrome P450 enzymatic activity, while ALT rises can be attributed to hepatocellular damage [10,12]. In the current study, only dogs given 4 mg/kg CBD/CBDA-rich hemp every 12 h showed significant alterations in ALP with 3/16 dogs showing elevations above the reference range. All other serum chemistry alterations were related to the time or sequence of treatment which may be related to the new environment or differences in feeding from their prior shelter or feral status. These rises in ALP appear to be innocuous as they are not accompanied by other serum hepatic enzyme alterations and prior investigations have shown that these rises are reversable when treatment is discontinued [14]. While several of the dogs were on concurrent trazodone for ease of handling for teaching purposes, it should be noted there was no evidence of neurologic deficits during co-administration of these treatments. Nor were there any adverse events or behavioral alteration noted by the student evaluation during co-treatment or even single-agent delivery of trazodone. Similarly, a report on the behavioral alterations in dogs when specifically co-administered trazodone at 10 mg/kg along with 1.4 mg/kg of CBD isolate as a single dose showed no behavioral alterations during co-administration or with each as single-agent therapy [17].

The lack of any behavioral or adverse events with the 2 mg/kg and 4 mg/kg every 12 h dosing regimen was surprising; however, the dogs used in this study were all under the estimated age of six years old and healthy with no comorbidities unlike clinical studies for epilepsy, arthritis and atopic dermatitis where there were concomitant treatments ongoing in dogs with a wider age range [4,7,8]. It must be noted that the veterinary evaluations were performed between 7:30 a.m. and video at 7 p.m. generally 30–60 min after morning dosing at 7 a.m. and 6 p.m. in the evening. Any anxiolytic or adverse events may peak at 1–2 h after dosing; therefore, our evaluations were not carried out just prior to peak serum exposure, similar to Hunt and colleagues [18]. Student assessments were typically performed throughout the day between 10 a.m. and 4 p.m., presenting a major limitation in our study being the lack of a standard timing for these evaluations, which was not possible due to the varied student activities throughout the day during this study. It must also be noted that this was not a validated survey and the variability in each student’s assessment could be related to timing; however, there were no real adverse events related to lethargy or somnolence and globally the dogs did not show any significant behavior changes during this chronic exposure. Though speculative, we might consider the student observations to be similar to what owners might report. Based on this information of no perceived differences between the treatment groups of placebo, 2 mg/kg and 4 mg/kg, it could be suggested that the administration of up to 4 mg/kg of this full-spectrum hemp product should not cause any significant adverse events or positive/negative behavioral effects in normal healthy dogs.

The management of stress and stress-related co-morbidities in canines has a limited number of viable pharmaceutical treatment options. In human psycho-therapeutic medicine, CBD has been purported to benefit users who suffer from anxiety, depression, and PTSD [23]. Veterinary medicine has begun to examine the behavioral effect of CBD in areas such as aggression and anxiety. Recent studies by Hunt and colleagues have examined a single car ride as well as repeated car rides and separation anxiety in kenneled dogs given a single oral dose of 4 mg/kg CBD isolate showing a lack of traditional measures such as lip licking, cortisol or heart rate variability changes globally [18,24]. However, behaviorist video assessment using qualitative behavioral ratings suggested lower stress, sad, tense and uncomfortable behaviors during vehicle travel [18]. This proved to be inconsistent in a follow-up study when examining multiple car rides at a single 4 mg/kg dose 1–2 h before three different car rides over 3 months [24]. The inconsistency and lack of definitive effects globally suggest that a CBD isolate may only have mild effects on anxiety and dosing may need to be higher. As noted by Hunt et al., studies on clinically affected dogs rather than kenneled dogs with no known anxiety disorders are needed to evaluate efficacy with owner surveys and board-certified veterinary behaviorist evaluation in a clinical population [18,24].

Blood was drawn from each dog three to four hours after the morning dose of the last day of each treatment period to assess cannabinoids found in the serum after dosing to assess relative steady state. It was evident that CBDA was absorbed and retained approximately two-fold better than CBD with 2 mg/kg every 12 h, with serum concentrations being approximately half of what was found of both cannabinoids when dosed at 4 mg/kg every 12 h, as expected. Interestingly, of the other minor cannabinoids in the product, the acidic THCA and CBCA appear to be absorbed and retained at some level suggesting that acidic cannabinoids may be absorbed better and preferable to the neutral cannabinoids such as THC and CBD. This has been shown in multiple species including parrots, humans, horses, primates and rabbits [20,24,25,26,27,28]. As expected, low concentrations of THC were found in the bloodstream, being a potential psychotropic cannabinoid, with the highest concentration observed being 8.3 ng/mL in the serum. With no psychotropic effects observed at the 4 mg/kg every 12 h in any dog, this suggests that this is a safe dose free of adverse events for normal healthy dogs.

As with any study, there were limitations to this study and in the interpretation of the data. The cohort utilized here were relatively young dogs and therefore may have different metabolic capacity than older dogs. Future studies utilizing these higher doses and potentially even higher doses in geriatric dogs (the typical cohort utilizing such products) is needed to fully assess the safety. This study only examined a short 2-week dosing regimen which may not be long enough to fully evaluate safety, similar to other studies showing safety in chronic dosing [11,12,13,14]; therefore, we cannot say what longer dosing regimens may reveal at 4 mg/kg. Lastly, as healthy dogs without any other significant medications on board, we cannot suggest that the use of this higher dose would necessarily be safe with other concomitant medications, although CBD and CBD-rich hemp products appeared to be safe when given with lower doses alongside non-steroidal anti-inflammatories and anti-epileptic drugs in 1–3 month clinical studies [4,6,8,9,10].

## 5. Conclusions

In this cohort of apparently healthy, young dogs, no adverse events were reported with the utilization of CBD/CBDA-rich hemp at 2 mg/kg and 4 mg/kg every 12 h. Additionally, at 2 mg/kg every 12 h, serum chemistry results had no significant findings and at 4 mg/kg every 12 h, a small portion of the dogs (3/16) had an elevation in ALP above the reference range, with no corresponding ALT elevation. Furthermore, even with concurrent use of trazodone in some dogs, hepatic injury was not apparent in the serum chemistry results. These findings, no adverse reactions/events and an innocuous rise in ALP, indicate that there is an adequate margin of safety in the use of CBD/CBDA-rich hemp extract for sub-acute use in dogs, though the use of a larger population of apparently healthy dogs with a wider range of ages would be prudent in future long-term studies utilizing CBD/CBDA-rich hemp at the higher dose. Regarding anxiety behavior, this study suggests minimal to no effect on behavior as observed by non-experts. Further evaluation utilizing board-certified behaviorists and possibly higher concentrations of cannabinoid could further define the necessary dose for anxiolytic effect.

## Figures and Tables

**Figure 1 animals-14-01863-f001:**
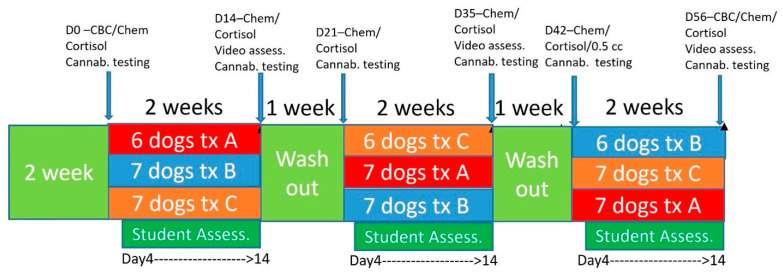
Timeline for treatment and procedures during the trial and treatment groups. Treatment A was 2 mg/kg BID, Treatment B was placebo, Treatment C was 4 mg/kg.

**Figure 2 animals-14-01863-f002:**
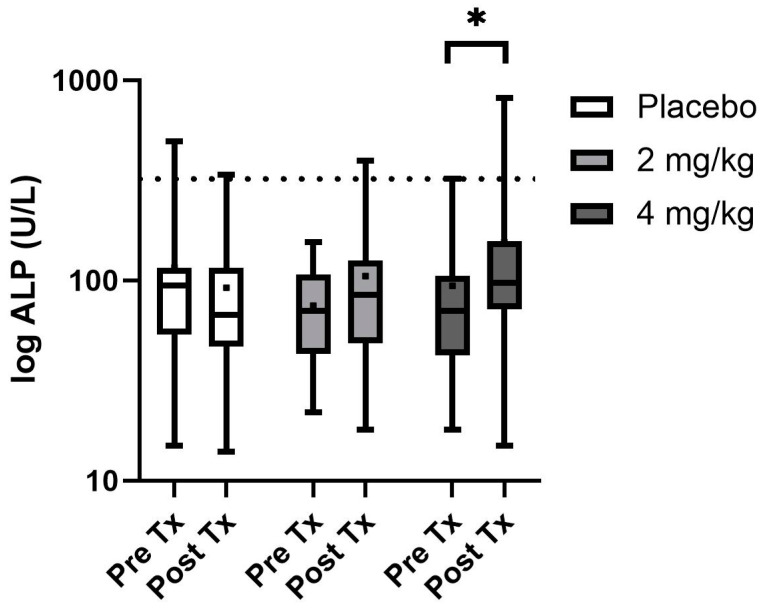
Serum ALP box and whisker plot with boxes showing the 25th and 75th median quartiles and whiskers indicating the minimum and maximum concentrations (dot within the box is mean) before treatment (day 0) and post-treatment (day 14) for each phase. Dotted line depicts the upper reference range. * Depicts a significant difference between the groups pre- and post-treatment.

**Table 1 animals-14-01863-t001:** Canine demographics of the cohort that completed the 10-week trial with assumed ages and weights during the trial.

Breed	Est. Age	Sex	Weight D0 (kg)	Weight D20 (kg)	Weight D41 (kg)
Pit mix	2 y	MN	33.2	34.0	33.6
Pit mix	2 y	FS	20.5	20.9	20
Lab Mix	2 y	FS	19.5	19.5	19.5
GSD Mix	2 y	MN	34.1	34.5	34.5
Border Collie Mix	7 m	FS	13.6	13.6	15
Great Dane mix	4 y	MN	21.8	27.3	32.7
German Shepherd Mix	2 y	MN	29.1	30.5	30
Shar pei mix	6 y	MN	26.4	26.4	25.9
Great Dane Mix	1 y	FS	28.6	29.5	30.4
Pit bull mix	2 y	FS	23.2	22.7	22.7
Pit bull mix	2 y	FS	22.7	22.7	22.7
Pit bull mix	2 y	MN	22.7	22.7	22.7
Lab Mix	1 y	FS	20.9	20.5	20.5
Terrier	2 y	FS	21.4	21.8	21.8
Coonhound	2 y	FS	33.2	33.6	33.6
Boxer mix	2 y	MN	25.9	25.9	25
Papillon	6 m	MN	9.1	10.5	11.4
Pit mix	2 y	FS	20.5	20.5	20.5
Border Collie Mix	1 y	MN	14.5	15.0	15

Note: These animals were acquired from the local shelter with minimal to nonexistent historical information. The ages were estimated and may not reflect true age.

**Table 2 animals-14-01863-t002:** Change from baseline scoring of dogs during students’ weekly evaluations during protocol based on 5-point Likert scaling: −2—much worse, −1—a little worse, 0—no change, 1—a little better, 2—a lot better.

Δ From Baseline	Placebo	2 mg/kg	4 mg/kg	Period	Tx	Seq	Gender	Traz
Anxiety	0.53 ± 0.58	0.48 ± 0.26	0.66 ± 0.56	0.47	0.27	0.10	0.36	0.81
Lethargy	0.46 ± 0.34	0.38 ± 0.37	0.50 ± 0.39	0.66	0.48	0.93	0.84	0.96
Handling	0.73 ± 0.58	0.63 ± 0.44	0.77 ± 0.59 *	0.70	0.008	0.04	0.3	0.89
Jumping	0.28 ± 0.48	0.14 ± 0.28	0.28 ± 0.34	0.47	0.41	0.96	0.76	0.14
Biting/mouthing	0.35 ± 0.52	0.20 ± 0.44	0.54 ± 0.52	0.20	0.08	0.57	0.79	0.14
Vocalizing	0.22 ± 0.42	0.16 ± 0.44	0.50 ± 0.49	0.02	0.19	0.18	0.43	0.07

* Asterisk indicates a significant difference between 4 mg/kg and 2 mg/kg treatment.

**Table 3 animals-14-01863-t003:** Means and standard deviations of CBC parameters (*n* = 16) at the beginning and end of the study period. Time, gender, and sequence significances are reported with *p* < 0.05 being significant and bolded.

Analyte (Ref. Range)	Pre Trial	Post Trial	Time	Gender	Tx Seq
Red blood cells (5.39–8.70 M/μL)	6.88 ± 0.79	7.14 ± 0.52	0.06	0.21	0.41
Hematocrit (38.3–56.6%)	46.8 ± 5.2	49.7 ± 5.1	**0.004**	0.08	0.5
Hemoglobin (13.4–20.7 g/dL)	15.7 ± 1.9	16.6 ± 1.5	**0.003**	0.08	0.47
Reticulocyte (10–110 K/μL)	70 ± 22	43 ± 15	**<0.001**	0.54	0.32
White blood cells (4.9–17.6 thous/μL)	14.1 ± 4.6	11.4 ± 3.6	**0.02**	0.78	0.43
Neutrophil (2.94–1270 thous/μL)	9.4 ± 4.0	6.7 ± 2.2	**0.015**	0.55	0.46
Lymphocytes (1.06–4.95 thous/μL)	3.28 ± 1.36	3.21 ± 1.49	0.99	0.22	0.83
Monocytes (0.13–1.15 thous/μL)	0.63 ± 0.23	0.55 ± 0.45	0.48	0.82	0.3
Eosinophils (0.07–1.49 thous/μL)	0.74 ± 0.45	0.93 ± 0.48	0.09	0.37	0.4
Basophils (0–0.1 thous/μL)	0.05 ± 0.03	0.02 ± 0.02	**0.015**	0.31	0.63
Platelets (143–448 thous/μL)	270 ± 114	230 ± 78	0.17	0.18	0.54

**Table 4 animals-14-01863-t004:** Mean and standard deviations of serum chemistry parameters (*n* = 16) at the beginning and end of the study period. Time, treatment (Tx), Tx*time, gender and sequence significances are reported, with *p* < 0.05 being significant.

Analyte (Ref. Range)	Time	Placebo	2 mg/kg	4 mg/kg	Tx	Time	Tx*Time	Gender	seq
Glucose	Pre	83 ± 11	79 ± 12	82 ± 8	0.37	<0.001	0.14	0.69	0.58
(53–114 mg/dL)	Post	76 ± 13	76 ± 11	70 ± 15					
SDMA	Pre	11 ± 1	12 ± 3	11 ± 2	0.91	0.15	0.82	0.2	0.14
(0–14 ug/dL)	Post	10 ± 3	11 ± 2	11 ± 2					
Creatinine	Pre	1.1 ± 0.2	1.0 ± 0.3	1.0 ± 0.2	0.79	0.1	0.43	0.94	0.06
(0.5–1.5 mg/dL)	Post	1.1 ± 0.3	1.1 ± 0.2	1.0 ± 0.1					
SUN	Pre	22 ± 7	22 ± 7	19 ± 6	0.25	0.24	0.86	0.75	0.06
(9–31 mg/dL)	Post	23 ± 9	22 ± 5	23 ± 5					
Phosphorus	Pre	6 ± 2	5.8 ± 1.3	5.9 ± 1.5	0.21	0.02	0.08	0.57	0.57
(2.5–6.1 mg/dL)	Post	5.2 ± 1	5.6 ± 1.4	5.9 ± 1.3					
Calcium	Pre	10.0 ± 0.4	9.9 ± 0.8	9.9 ± 0.5	0.34	<0.001	0.65	0.62	0.91
(8.4–11.8 mg/dL)	Post	9.4 ± 0.5	9.4 ± 0.6	9.7 ± 0.6					
Sodium	Pre	147 ± 4	144 ± 6	147 ± 2	0.14	0.76	0.41	0.23	0.75
(142–152 mmol/L)	Post	144 ± 9	146 ± 3	148 ± 3					
Potassium	Pre	4.6 ± 0.4	4.5 ± 0.3	4.6 ± 0.3	0.1	<0.001	0.43	0.09	0.89
(4.0–5.4 mmol/L)	Post	4.7 ± 0.4	4.7 ± 0.4	4.9 ± 0.5					
Chloride	Pre	110 ± 3	107 ± 5	110 ± 2	0.06	0.11	0.06	0.76	0.58
(108–119 mmol/L)	Post	110 ± 2	110 ± 2	110 ± 2					
Total protein	Pre	6.1 ± 0.5	6.0 ± 0.4	6.0 ± 0.5	0.77	0.18	0.3	0.02	0.27
(5.5–7.5 g/L)	Post	6.0 ± 0.4	5.9 ± 0.4	6.1 ± 0.5					
Albumin	Pre	3.0 ± 0.3	3.0 ± 0.3	3.0 ± 0.3	0.88	0.96	0.76	0.01	0.73
(2.7–3.9 g/L)	Post	3.0 ± 0.3	3.0 ± 0.3	3.1 ± 0.4					
Globulin	Pre	3.1 ± 0.4	3.0 ± 0.4	3.0 ± 0.4	0.73	0.07	0.12	0.55	0.4
(2.4–4.0 g/L)	Post	3.0 ± 0.4	2.9 ± 0.3	3.0 ± 0.4					
ALT	Pre	38 ± 11	40 ± 14	39 ± 9	0.66	0.95	0.49	0.19	0.01
(18–121 U/L)	Post	40 ± 10	42 ± 11	39 ± 8					
AST	Pre	29 ± 6	31 ± 8	31 ± 7	0.4	0.74	0.42	0.02	0.68
(16–55 U/L)	Post	30 ± 7	32 ± 9	30 ± 5					
ALP	Pre	116 ± 111	75 ± 40	94 ± 79	0.92	<0.001	<0.001	0.29	0.27
(5–160 U/L)	Post	92 ± 79	105 ± 88	155 ± 187 *					
Bilirubin	Pre	0.1 ± 0.1	0.1 ± 0.1	0.1 ± 0.1	0.98	0.29	0.07	0.37	0.002
(0–0.2 U/L)	Post	0.1 ± 0.1	0.1 ± 0.1	0.0 ± 0.0					
Cholesterol	Pre	232 ± 60	214 ± 52	228 ± 57	0.91	0.38	0.06	0.37	0.14
(131–345 mg/dL)	Post	221 ± 47	227 ± 47	227 ± 46					
Cortisol	Pre	1.3 ± 0.6	1.6 ± 1.1	1.9 ± 1.1	0.23	0.98	0.44	0.27	0.61
(0.6–3.4 μg/dL)	Post	1.4 ± 0.8	1.3 ± 0.6	2.2 ± 2.1					

* Asterisk indicates a significant difference between 4 mg/kg treatment from both placebo and 2 mg/kg treatment.

**Table 5 animals-14-01863-t005:** Mean (median; range) serum concentration of cannabinoids and CBD-related metabolites (*n* = 16) 3–4 h after morning dose on day 14 of 2 and 4 mg/kg BID dosing of CBD/CBDA-rich hemp extract.

Analyte	LLOD	LLOQ	2 mg/kg	4 mg/kg
CBD	5	10	37.9 (37.4; 10.5–82.8)	80.4 (80.7; 17.0–119.8)
CBDA	1	2.5	101.5 (90; 33.1–221.0)	189.7 (154.4; 35.6–674.6)
THC	1	2.5	1.5 (1.3; 0–5.0)	4.8 (5.3; 1.3–8.2)
THCA	1	2.5	6.3 (6.0; 1.3–13.6)	8.6 (7.8; 3.1–14.7)
CBCA	1	2.5	16 (25.6; 4.3–30.2)	24.3 (20.2; 7.3–51.7)
6-OH CBD	1	5	LLOQ	4.2 (3.9; 2.5–6.9)
7-COOH-CBD	0.5	1	LLOQ	1.1 (1.2; 0.5–2.3)

LLOQ indicates values were below the lower limit of quantification.

## Data Availability

The original contributions presented in the study are included in the Article/Appendix A; further inquiries can be directed to the corresponding author/s.

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
