# Peer review of "Tolerability of 2 and 4 mg/kg Dosing Every 12 Hour of a Cannabidiol- and Cannabidiolic Acid-Rich Hemp Extract on Mixed-Breed Dogs Utilized for Teaching in a Closed Colony"

_animals, 2024, doi:10.3390/ani14131863_

Round 1

Reviewer 1 Report

Comments and Suggestions for Authors

Author Response

Overall this is a well wri?en manuscript and an interes?ng topic. The following are some minor

comments/suggested edits.

AU:  Thank you for the positive comments on this manuscript.  All of the revisions required are in red in the revised manuscript for easy recognition

  1. Line 73 – first word “u?lize” should be changed to “u?lized”.

AU: Has been changed

  1. Line 99 – Sunflower oil was used as the placebo, can a li?le more detail be provided to be sure

the oil was appropriate food grade oil.

AU:  Yes we have clarified this in the manuscript as this is the same sunflower oil used in the current commercial preparation

  1. Line 111 – please elaborate how the 1 week washout period was determined to be adequate.

AU:  We proposed assumed adequate washout period from prior studies conducted as we were looking for acute effects and the minor residual CBD would not conflicted with the tolerability results or the measured cannabinoids significantly at the end of the study.  In addition we did measure all serum samples at the end of the phase and just before the phase and found no cannabinoids – which we have also stated in the manuscript.

  1. Line 122 – how were these students qualified to conduct these clinical examina?ons – I assume

they were students in a veterinary program but this is not clear.

AU:  We have clarified that the students were examining only for behavioral and adverse events as if owners, while the physical examinations were performed by veterinarians

  1. Were students blinded to the current treatment when they performed the evalua?ons?

AU:  Yes the students were blinded as well.  This has been clarified

  1. Line 166 – please clarify how frequently this serum was collected and frozen for the serum

cannabinoid analysis. I assume it was at the end of each treatment period based on Figure 1,

but this is not very clear in the text.

AU: We have clarified that an aliquot of serum from the chemistry screen at the end of the treatment phase was reserved for cannabinoid analysis

  1. Line 236 and 237 – please expand on what parameters were evaluated in June the physical,

neurologic and orthopedic examina?ons, this can be in supplementary data.

AU:  We have added in parentheses in each case the exact parameters that were assessed for neurological and orthopedic examinations

  1. Discussion sec?on, I appreciate the extent to which the authors discussed the limita?ons of this

study which are not insignificant. In this sec?on (lines 430 to 433), I find the last sentence

confusing, can it be reworded so the intent is clearer?

AU:  This has been reworded as such for clarity.

  1. Line 471 – I think the word “doses” is missing from between “even higher in” and “geriatric”.

AU: yes – have clarified with doses in each area of the manuscript

  1. Line 472 – suggest removing the word “chronic” from the sentence “a short 2-week chroni

dosing..” – 2 weeks is not chronic dosing. 

AU:  We have removed the word chronic as suggested

  1. Please ensure it is clear throughout the document if the animals received 2 mg/kg bw or 4

mg/kg bw every 12 hours or if this was the total daily dose, given in divided doses. As an

example, in lines 430 to 431 the authors state “…that dosing between 2 and 4 mg/kg should

….”but then in line 465 the authors state ..”at 4 mg/kg every 12 hours…”. These are only 2

examples but clarity is needed throughout the manuscript.

AU:  Yes this can be confusing so we have added where dosing was every 24 hours vs every 12 hours throught the manuscript when comparing to other safety studies.

Reviewer 2 Report

Comments and Suggestions for Authors

Thank you for the opportunity to review this manuscript on the use of two different doses of CBD/CBDA rich hemp extract in a population of healthy research dogs. The study design is appropriate and adequately outlined such that the study could be readily replicated with the information provided. The statistical analyses appear appropriate for the data collected. Results are clear and the discussion is thorough. 

I identified a number of typographical errors and just a few areas where I would appreciate clarification or additional information to be provided: 

Title: Consider editing title from “increasing dose”, as I was initially expecting a different study design based on this wording (multiple dose escalations) instead of just assessing 2 different doses.

Line 39: Capitalization missing for United States 

Line 70-77: Please provide your hypotheses. For the first aim, it is my understanding that you were assessing both adverse effects of the CBD/CBDA dosing used (not currently written as part of the aim) as well as adverse events/behavior related to kenneled lifestyle. Can you please edit this first aim to reflect this? 

Line 73: Typo, “utilize” should be “utilized” 

Line 83: “acclimatization” should be “acclimation” here unless you specifically mean the dogs were allowed to adjust to the climate of their new environment 

Would you be able to more clearly define your use of “period” and “sequence”?  

Line 114: Typo, “concentrations” is written twice 

Line 118: Who performed the weekly physical examinations?  

Line 145: Typo, “though” should be “through” 

Line 174: Typo, “generating” should be “generated” 

Line 238-239: Should these be 23.7 +/- 6.7 kg and 24.1 /- 6.6 kg, missing the minus sign? 

Table 1: You previously indicated that unsterilized dogs that entered the trial were spayed and neutered during the acclimation period before D0, but this is not reflected in the table. Also, can weight please be converted to kilograms please? 

Line 250 and 268: Typo, “trazadone” should be “trazodone” 

Line 253: Typo “5mg/kg every q hours trazodone” please correct for how frequently trazodone was administered 

Line 324 and Figure 1: I may be misunderstanding here, but based on Figure 1 I don’t see a group of dogs who received treatment series in the order of CAB. Should this be CBA? 

Line 367: Typo “attributed to the use treatment” 

Line 431: Typo “due to the lack no perceive differences” 

Line 448: Typo, “effected” should be “affected” 

Line 465-466: I would specify that this appeared to be a safe dose of THC free of adverse events for normal healthy dogs 

Line 468: Typo “limitation” should be “limitations”

Author Response

Thank you for the opportunity to review this manuscript on the use of two different doses of CBD/CBDA rich hemp extract in a population of healthy research dogs. The study design is appropriate and adequately outlined such that the study could be readily replicated with the information provided. The statistical analyses appear appropriate for the data collected. Results are clear and the discussion is thorough. 

I identified a number of typographical errors and just a few areas where I would appreciate clarification or additional information to be provided: 

AU: Thank your for the thorough review of this manuscript and the suggestions pointed out below – all of the changes are in red for better identification throughout the manuscript for your approval

Title: Consider editing title from “increasing dose”, as I was initially expecting a different study design based on this wording (multiple dose escalations) instead of just assessing 2 different doses.

AU: We have changed this to 2 mg/kg and 4 mg/kg every 12 hour dosing for clarity

Line 39: Capitalization missing for United States 

AU: has been changed

Line 70-77: Please provide your hypotheses. For the first aim, it is my understanding that you were assessing both adverse effects of the CBD/CBDA dosing used (not currently written as part of the aim) as well as adverse events/behavior related to kenneled lifestyle. Can you please edit this first aim to reflect this? 

AU: A hypothesis has been added and aim one has been clarified as such

We hypothesized that increasing dosing from 2 mg/kg every 12 hours to 4 mg/kg every 12 hours would elicit adverse effects, behavioral alterations resulting in tolerability issues. The aim of our study was to examine adverse events at two different doses of CBD/CBDA rich hemp within the dosing range utilize in multiple studies using medical records and veterinary examination weekly for adverse events; as well as student evaluation of the dogs in an “owner like” unvalidated observation related to their kenneled lifestyle

Line 73: Typo, “utilize” should be “utilized” 

AU: has been changed

Line 83: “acclimatization” should be “acclimation” here unless you specifically mean the dogs were allowed to adjust to the climate of their new environment 

AU: Thank you this has been changed

Would you be able to more clearly define your use of “period” and “sequence”?  

AU: This has been clarified as such  in the text “washout periods between treatment periods (two week duration) and each dog was assigned to a sequence of treatment (i.e. ACB, BAC, or CBA)

Line 114: Typo, “concentrations” is written twice 

AU; Has been removed.

Line 118: Who performed the weekly physical examinations?  

AU: has been clarified as the veterinary PI

Line 145: Typo, “though” should be “through” 

AU; Has been changed

Line 174: Typo, “generating” should be “generated” 

AU: has been changed

Line 238-239: Should these be 23.7 +/- 6.7 kg and 24.1 /- 6.6 kg, missing the minus sign? 

AU: Yes – the minus signed have been added as underlines of the +

Table 1: You previously indicated that unsterilized dogs that entered the trial were spayed and neutered during the acclimation period before D0, but this is not reflected in the table. Also, can weight please be converted to kilograms please? 

AU: The tables has been revised accordingly

Line 250 and 268: Typo, “trazadone” should be “trazodone” 

AU: has been changed

Line 253: Typo “5mg/kg every q hours trazodone” please correct for how frequently trazodone was administered 

AU: has been changed

Line 324 and Figure 1: I may be misunderstanding here, but based on Figure 1 I don’t see a group of dogs who received treatment series in the order of CAB. Should this be CBA? 

AU:  Yes this is a typo on our part it is the CBA sequence group and has been changed – thank you for catching this. 

Line 367: Typo “attributed to the use treatment” 

AU: Use has been taken out

Line 431: Typo “due to the lack no perceive differences” 

AU: Changed to …lack of perceived differences

Line 448: Typo, “effected” should be “affected” 

AU: has been changed

Line 465-466: I would specify that this appeared to be a safe dose of THC free of adverse events for normal healthy dogs 

AU: Yes point taken normal healthy has been added

Line 468: Typo “limitation” should be “limitations”

AU: has been added